# Pseudohyperaldosteronism Due to Licorice: A Practice-Based Learning from a Case Series

**DOI:** 10.3390/ijms25137454

**Published:** 2024-07-07

**Authors:** Chiara Sabbadin, Andrea Graziani, Alessandro Bavaresco, Pierluigi Mazzeo, Irene Tizianel, Filippo Ceccato, Decio Armanini, Mattia Barbot

**Affiliations:** 1Unit of Endocrinology, University Hospital of Padua, 35128 Padua, Italy; agraziani1995@gmail.com (A.G.); alessandrobavaresco@gmail.com (A.B.); piermazzeo94@gmail.com (P.M.); irenetizianel@gmail.com (I.T.); filippo.ceccato@unipd.it (F.C.); mattia.barbot@unipd.it (M.B.); 2Department of Medicine (DIMED), University of Padua, 35128 Padua, Italy; decio.armanini@unipd.it

**Keywords:** pseudohyperaldosteronism, licorice, hypokalemia, hypertension, 11-hydroxysteroid dehydrogenase

## Abstract

Pseudohyperaldosteronism (PHA) is characterized by hypertension, hypokalemia, and a decrease in plasma renin and aldosterone levels. It can be caused by several causes, but the most frequent is due to excess intake of licorice. The effect is mediated by the active metabolite of licorice, glycyrrhetinic acid (GA), which acts by blocking the 11-hydroxysteroid dehydrogenase type 2 and binding to the mineralocorticoid receptor (MR) as an agonist. The management of licorice-induced PHA depends on several individual factors, such as age, gender, comorbidities, duration and amount of licorice intake, and metabolism. The clinical picture usually reverts upon licorice withdrawal, but sometimes mineralocorticoid-like effects can be critical and persist for several weeks, requiring treatment with MR blockers and potassium supplements. Through this case series of licorice-induced PHA, we aim to increase awareness about exogenous PHA, and the possible risk associated with excess intake of licorice. An accurate history is mandatory in patients with hypertension and hypokalemia to avoid unnecessary testing. GA is a component of several products, such as candies, breath fresheners, beverages, tobacco, cosmetics, and laxatives. In recent years, the mechanisms of action of licorice and its active compounds have been better elucidated, suggesting its benefits in several clinical settings. Nevertheless, licorice should still be consumed with caution, considering that licorice-induced PHA is still an underestimated condition, and its intake should be avoided in patients with increased risk of licorice toxicity due to concomitant comorbidities or interfering drugs.

## 1. Introduction

Pseudohyperaldosteronism (PHA) is a medical condition characterized by hypokalemia and hypertension. It clinically mimics primary aldosteronism, but both renin and aldosterone levels are markedly reduced. PHA can occur due to endogenous causes, such as a deficiency of 17α-hydroxylase or 11β-hydroxylase, primary kidney channelopathy, activation or mutations of the mineralocorticoid receptor (MR), alterations of expression, or saturation of 11-hydroxysteroid dehydrogenase type 2 (11HSD2). PHA is also induced by exogenous causes, such as chronic intake of some corticosteroids, abiraterone, carbenoxolone, oral contraceptives, hypersodic diets, grapefruit abuse (which influences drug pharmacokinetic-reducing CYP3A4 activity), and licorice intake. Hypertension with low renin and aldosterone can also be related to particular causes of hypertension (such as sclerosis of the juxtaglomerular apparatus, aging, low-renin essential hypertension, and partial/total nephrectomy) [1,2].

Licorice is a perennial plant that has been studied for a long time for its multiple biological and endocrine properties [1,3]. Its international Latin name is Glycyrrhiza glabra, derived from the Greek words glykos (sweet) and rhiza (root). Glycyrrhizic acid (GL) and glycyrrhetinic acid (GA) are the most important active compounds of licorice. They have a broad spectrum of biological activities, especially anti-inflammatory, anti-androgen, anti-viral effects [4,5]. After oral ingestion, it is hydrolyzed to GA in the stomach and duodenum (Figure 1). GA presents extensive pharmacological activities, and it has long been utilized as a natural sweetener and thirst reliever. The yellow color of licorice is due to the flavonoid content of the plant, which includes glabridin that has an estrogen-like effect [6,7]. All these beneficial properties might induce people to licorice abuse, neglecting the potential side effects, such as hypertension [8]. This effect is predominantly due to GA, which acts through two different mechanisms: it can both bind the MR and block 11HSD2 activity [3]. Cortisol and aldosterone bind the MR with similar affinity, but cortisol levels are 100 to 1000 times higher than that of aldosterone. For this reason, in the classical target tissues of aldosterone, such as kidneys, intestines, and salivary and sweat glands, the MR is activated by aldosterone due to the presence of the enzyme 11HSD2 that converts active cortisol to locally inactive cortisone. However, if the enzyme is saturated or blocked, such as during prolonged or increased intake of licorice, cortisol can bind and activate the MR, leading to sodium reabsorption, potassium excretion, metabolic alkalosis, and hypertension [2]. Moreover, increased levels of GA directly bind the MR, notwithstanding the low affinity. It has been proven in human mononuclear leukocytes by radioreceptor assay that the coincubation with canrenone, a spironolactone metabolite, can reverse the mineralocorticoid effect of GA [9]. For this reason, MR blockers can be useful in licorice-induced PHA.

Other than PHA, excessive licorice intake has been associated with rhabdomyolysis, muscle paralysis, respiratory impairment, hypertensive emergencies, encephalopathy, and acute renal failure, probably related to marked hypokalemia [10]. The MR is expressed in vascular and inflammatory cells; its chronic activation in the case of PHA may also lead to systemic inflammation, endothelial dysfunction, and increased cardiovascular risk [2].

PHA can occur during prolonged intake not only of licorice roots but also of products flavored with licorice, cosmetics, both traditional and herbal medicine, and even laxatives [1,3]. The effects are reversible, but the time of recovery may require even 2–3 weeks after licorice withdrawal and a period with MR blockers to treat hypertension and hypokalemia [11].

Here, we present three cases of PHA due to various licorice intakes and with different clinical presentations. Informed consent was obtained from all individual participants included in the study.

Through this case series, we aim to increase awareness about exogenous PHA and the possible risk associated with excess intake of licorice, which often requires hospitalization and, therefore, represents a significant health concern. The pathogenetic, diagnostic, and therapeutic aspects will be discussed to aid physicians in the detection and management of this dangerous condition and in providing adequate information to their patients about the risk of licorice abuse.

## 2. Case Report 1

A 71-year-old man was admitted to our emergency room (ER) for severe lower limb hyposthenia. He presented elevated blood pressure (BP) values (190/80 mmHg), severe hypokalemia (1.6 mmol/L), metabolic alkalosis, and Troponin I (TnI) elevation (41 ng/L, normal value < 17). The electrocardiogram (ECG) showed a sinus rate of 88 beats per minute (bpm) and diffuse ST depression. His medical history was characterized by chronic obstructive pulmonary disease, diabetes mellitus treated with metformin, and hypertension treated with amlodipine 10 mg/day. The patient was transferred to the Intensive Cardiology Unit (ICU) and treated with a potassium and nitroprussiate continuous infusion. After 3 days, ECG abnormalities regressed, whereas hypertension and hypokalemia persisted, so the patient was transferred to our Endocrine Unit. The patient was still treated with potassium (40 mEq/day intravenous and 3600 mg/day orally) and magnesium supplementation (1 g/day), potassium canrenoate (100 mg/day), nitroprussiate infusion (2 cc/h), doxazosin (2 mg/day), and amlodipine (10 mg/day). Biochemical evaluation revealed normal 24 h urinary metanephrines and free cortisol (UFC), whereas orthostatic direct renin (DRC) and plasma aldosterone concentrations (PAC) were suppressed (<2 mIU/L and 41.2 pmol/L, respectively). When collecting the personal history, the patient reported an important intake of licorice candies, about 120 g/day, in the 2 weeks prior to hospitalization. A blood sample was collected to perform the analysis of GA using liquid chromatography–mass spectrometry, and the results showed a significant peak, corresponding to 5.78 ± 0.20 μg/mL (Figure 2). Thus, we diagnosed licorice-induced PHA. During the following days, potassium and nitroprussiate infusions were gradually stopped, while oral potassium supplements and anti-hypertensive drugs were maintained. On the fifth day of hospitalization, the patient developed high-ventricular rate atrial fibrillation, which was treated with anti-coagulation (apixaban) and beta-blockers without efficacy. Diltiazem therapy was later begun intravenously (7 cc/h) and then orally (180 mg/day), with continuous correction of potassium. Anti-hypertensive therapy was optimized by adding 5 mg/day of ramipril. The patient was then discharged with normalized potassium levels and multiple-drug therapy (apixaban, ramipril, diltiazem, potassium canrenoate, and his previous anti-diabetic therapy). Two weeks after discharge, the patient’s potassium and BP values were normal, and DRC and PAC were increased (32.54 mIU/L and 231 pmol/L, respectively). Thus, the MR blocker was stopped, and anti-hypertensive treatment was reduced.

## 3. Case Report 2

A 55-year-old man was admitted to our ER because of chest pain associated with fatigue, apathy, and paresthesia in the left upper limb for a week. He presented elevated BP values (160/90 mmHg), severe hypokalemia (2.2 mmol/L), and TnI elevation (38 ng/L). The ECG showed a sinus rate of 108 bpm, supraventricular extrasystoles, flattened T-wave, and left anterior hemiblock. He complained of a recent onset of hypertension treated with olmesartan (40 mg/daily) and amlodipine (5 mg/day). Urgent coronary computed tomography (CT) angiography excluded significant coronary stenosis. The patient was transferred to our Endocrine Unit and treated with low-dose aspirin, bisoprolol (2.5 mg/day), doxazosin (2 mg/day), and potassium supplements (60 mEq/day intravenous and 3600 mg/day orally). Because of persistent hypertension and hypokalemia, we increased both potassium supplements (120 mEq/day intravenous and 5400 mg/day orally) and doxazosin dosage (6 mg/day) and added potassium canrenoate (100 mg twice a day) and amlodipine (5 mg/day). Screening for secondary hypertension was performed despite interfering with concomitant medications. While waiting for hormone testing, a direct abdominal CT was performed, which excluded adrenal lesions. Biochemical exams revealed normal 24 h urinary metanephrines and UFC but increased urinary cortisol to cortisone ratio (UFC 283 nmol/24 h, urinary cortisone [UEC] 133 nmol/24 h, UF/E ratio 2.12, normal value 0.14–1.08). DRC was suppressed, and PAC was under the lower normal limit (<2 mIU/L and 63.2 pmol/L, respectively). The patient first denied the consumption of licorice, grapefruit juice, or glucocorticoid therapy, but he then recalled the habit of drinking licorice liqueur at bedtime every day.

The patient was discharged with a normal potassium level but was still hypertensive despite taking multiple anti-hypertensive drugs (doxazosin, potassium canrenoate, amlodipine). After 2 weeks, the patient’s BP values were normalized, and biochemical exams showed moderate hyperkalemia, normalization of UF/E ratio (0.9), and a progressive increase of DRC and PAC (7.8 mIU/L and 95.4 pmol/L, respectively). Thus, the MR blocker was stopped.

## 4. Case Report 3

A 52-year-old woman was referred to our Endocrinology Unit for the recent onset of fluid retention and unexplained weight gain (4 kg in 3 months). On biochemical exams, thyroid function, electrolytes, 24 h urinary metanephrines, and UFC were all normal, whereas DRC and PAC were suppressed (<2 mIU/L and 63.2 pmol/L, respectively). She had no familial or personal history of hypertension or diabetes. She entered menopause at 50 years of age. She denied the consumption of drugs, licorice, or grapefruit juice. The patient reported having started a supplement with bromelain a few days before to address the fluid retention, but without efficacy. She had been using an anti-wrinkle face cream for several months. On examination, she had moderate hypertension (140/90 mmHg) and mild fluid retention on the legs and on the arms. Suspecting PHA due to some ingredients in the cream, we suggested stopping both the supplement and the cream and starting doxazosin (2 mg/daily). After 4 weeks, fluid retention and weight were improved. The exams revealed the normalization of DRC and PAC. The patient brought the label of the face cream, and stearyl glycyrrhetinate was one of the ingredients (a full list is reported in Table 1), supporting the diagnosis of a licorice-induced PHA.

## 5. Discussion

Licorice-induced PHA is one of the most common causes of exogenous hypertension, but it is still an underestimated condition for several reasons. Firstly, there are many licorice-containing products that are readily available for our everyday use, such as candies, breath fresheners, teas, liquors, tobacco, cosmetics, and laxatives [1,3]. Their consumption is considered a natural remedy with health benefits, but the potential adverse effects due to overconsumption are quite underestimated in the general opinion [10]. Secondly, it is difficult to establish the unsafe level of licorice consumption because the GA content can vary markedly between the products containing licorice [12]. A study published in 1994 demonstrated that a daily oral intake of 1–5 g of licorice is safe in most healthy adults [13]. Two years later, in a double-blinded randomized placebo-controlled study with a larger sample of volunteers and a longer period of exposure to different doses of pure GL, the authors reported that 2 mg/kg/day of GL was not associated with any adverse effects [14]. At present, the toxic threshold of GL defined by the World Health Organization is 100 mg/day. Finally, the toxic effects are influenced not only by the amount of licorice intake but also by several individual factors, such as long-term use, female sex, aging, constipation, associated drugs used (as oral contraceptives, sympathomimetic agents, non-steroidal anti-inflammatory drugs, and steroids), and comorbidities (as impaired cardiac, renal or liver function, and hypertension) [15,16].

For example, females are more susceptible to the adverse effects of licorice. This could be due to the prolonged gastrointestinal transit time of women, affecting the hydrolyzation of GL into GA, and to the estrogenic and anti-androgenic activity of licorice, which modulates the effects on BP values. A previous study reported lower serum aldosterone levels in men than in women after the administration of the same dose of licorice for the same period of time [17]. A recent review of case reports of licorice toxicity detected a slight male prevalence (57 out of 104) but an increased frequency of adverse cardiac events in females (10 out of 13) [18].

Even the elderly may be more predisposed to developing licorice-induced PHA due to the presence of other comorbidities and the age-dependent decrease of 11HSD2 function [19]. On the contrary, licorice toxicity is an extremely rare condition among children, with only two cases reported so far [20,21].

Another factor that might influence the onset of PHA is the presence of 11HSD2 polymorphisms. Several mutations of the gene encoding 11HSD2 have been identified and associated with the apparent mineralocorticoid excess (AME) [22]. This is an autosomal recessive disorder responsible for a rare cause of PHA, characterized by juvenile-resistant low-renin hypertension, marked hypokalemic alkalosis, low aldosterone levels, and high ratios of cortisol to cortisone metabolites. Even other molecular mechanisms may alter 11HSD2 activity, influencing the development of low-renin or salt-sensitive essential hypertension or the individual susceptibility to GA.

Considering all these influencing conditions, licorice intake has been associated with a wide range of clinical situations. Hypertension and hypokalemia are the most frequent features. As previously explained, these effects are due to GA, which binds the MR and blocks 11HSD2 activity [3]. A recent meta-analysis reported a significant correlation between daily dose of GL and BP values [12]; the net change in BP and potassium levels was not relevant, but it might be for subjects with pre-existing cardiovascular diseases.

As reported in our first clinical case, an increased acute intake of licorice may lead to uncontrolled hypertension and acute hypokalemia, which could cause cardiac adverse events in patients with cardiovascular risk factors. Hypokalemia is associated with an increased risk of ECG alterations, arrhythmia, hospitalization, and cardiac arrest [23]. Moreover, recent studies show that patients with low potassium levels, especially below 3.5 mmol/L, present a higher risk of developing atrial fibrillation [24,25], as happened in our first case. The concomitant intake of loop diuretics and thiazides may enhance hypokalemia and increase the risk of cardiotoxicity, especially in patients treated with digoxin. Therefore, in patients taking these medicines, licorice intake should be completely avoided. The prognosis is generally good, with complete symptom resolution after 2–3 weeks after licorice suspension in most of the cases, but some patients can show persistent mineralocorticoid-like effects. This could be due to large volume of GA distribution, its long half-life (12 h), prolonged enterohepatic circulation, and other complex hormonal effects exhibited by the different compounds of licorice in addition to its mineralocorticoid-like activity [17,26].

Since a direct antidote does not exist, the treatment consists of stopping the licorice intake, correcting the hydro-electrolyte imbalance, and reducing BP values, especially with MR blockers, to avoid severe complications [11]. Indeed, other cardiovascular disorders may be induced by licorice intake: hypertensive encephalopathy and retinopathy, pulmonary edema, and acute kidney injury are commonly reported [18].

Our second case is another typical example of licorice-induced PHA. Since the patient was 55 years old and had a recent onset of resistant hypertension with marked hypokalemia, the clinical picture could be suggestive of endocrine hypertension. It is mandatory to perform an accurate history and investigate the patient’s lifestyle and habits, including questions about water and sodium intake and the use of licorice, grapefruit juice, contraceptives, or glucocorticoids, to avoid the prescription of unnecessary exams, such as the morphological evaluation of adrenals. There are many beverages that contain licorice among their ingredients. In literature, the most frequent cases of licorice toxicity are associated with the daily assumption of herbal tea containing licorice [27]. The daily use of licorice liqueur is less common, and the amount of licorice may vary. However, chronic daily intake, even as a digestif, could be sufficient to induce PHA, as happened in our second patient.

Biochemical exams should always include the evaluation of renin, aldosterone, cortisol, sodium, and potassium, considering possible interfering drugs, such as diuretics, ACE-inhibitors, sartanics, MR-antagonists, beta-blockers, corticosteroids, and combined contraceptives [3]. The finding of low levels of aldosterone and renin is suggestive of PHA. Some particular measurements, such as the increase of UF/E ratio or their reduced metabolites (tetrahydrocortisol + allo-tetrahydrocortisol and tetrahydrocortisone, respectively), may confirm the reduced function of 11HSD2 [28], as happened in our second case. However, this analysis is not available in the majority of diagnostic laboratories, and only the history and the clinical/biochemical examination may differentiate AME syndrome from exogenous causes that block 11HSD2, such as licorice, grapefruit, and carbenoxolone (a GA derivate, used for the treatment of gastrointestinal ulcerations and inflammation) [3]. The determination of the suspected exogenous substance with spectrometry or chromatography analysis could be another useful diagnostic tool, as reported in our first clinical case. However, even this analysis is not very common. Therefore, in patients with a suspected exogenous PHA, the diagnosis is confirmed with the clinical resolution and the normalization of aldosterone and renin levels after discontinuation of the substance.

Many other clinical situations have been associated with licorice intake, probably related to marked hypokalemia, such as muscular, neurological, and gastrointestinal disorders [18]. Some patients could complain only of water retention and moderate hypertension, without electrolyte and renal abnormalities, as reported in our third case. This is a typical feature of “licorice syndrome”: in fact, the occurrence of edema is in contrast to mineralocorticoid excess, where edema is typically absent for the “sodium escape” phenomenon [26]. After long-term ingestion of licorice, plasma levels of anti-diuretic hormone, renin, and aldosterone decreased, while the concentration of atrial natriuretic peptide increased as a physiological, albeit ineffective, response to prevent fluid retention and development of hypertension [29]. Considering the recently published revision of cases of licorice-induced PHA [18], our third patient is the first reported case of PHA that was induced by a cosmetic product containing a derivate of licorice. Stearyl glycyrrhetinate is another derivative of licorice that is used in the cosmetic industry due to its skin-soothing and mattifying properties [30]. Like GL and GA, it could have some mineralocorticoid-like effect, as suggested by the clinical and biochemical resolution after discontinuation of the cream.

The main limitation of these case reports is the limited possibility of generalizing the validity of the studies. However, the specific measurements of licorice compounds using spectrometry or chromatography analysis and the temporary alteration of UF/E ratio, as performed in the first two cases, confirm the cause–effect relationship between licorice intake and PHA.

## 6. Conclusions

In recent years, the mechanisms of action of licorice and its active compounds have been better elucidated, suggesting its use in several clinical settings as an anti-inflammatory, anti-viral, anti-androgen, and even anti-cancer agent. Nevertheless, licorice should still be consumed with caution, considering that licorice-induced PHA is still an underestimated condition. Intake of licorice and licorice-derived products, including drinks, tobacco, laxatives, candies, and even cosmetics, should always be suspected and excluded in the presence of hypertension, hypokalemia, and low concentrations of plasma renin and aldosterone. Considering all the other causes of PHA, the association of licorice intake could lead to a more severe picture in females, elderly people, and those treated with oral contraceptives, steroids, or non-potassium-sparing diuretics. In these groups, licorice intake should be completely avoided. This case series aims to increase awareness among physicians about exogenous PHA. We hope that this message could also be transmitted from doctors to their patients to change popular beliefs and dietary habits that could be dangerous.

## Figures and Tables

**Figure 1 ijms-25-07454-f001:**
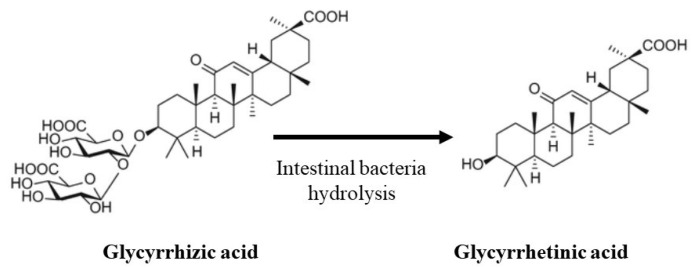
Chemical structure of glycyrrhizic acid and glycyrrhetinic acid.

**Figure 2 ijms-25-07454-f002:**
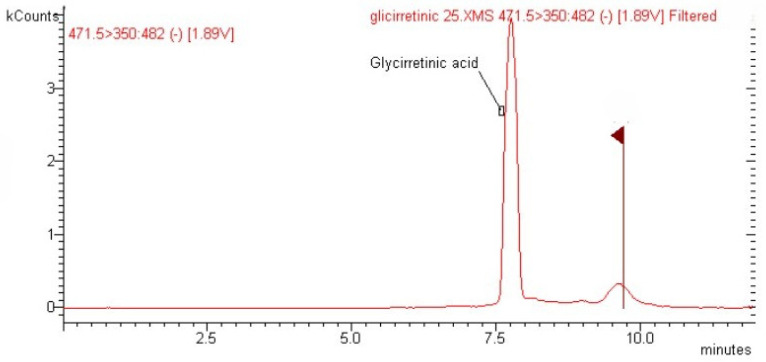
Liquid chromatography–mass spectrometry (LC-MS) analysis of glycyrrhetinic acid (*m*/*z* 471) in the blood sample of patient 1. The concentration of glycyrrhetinic acid corresponded to 5.78 ± 0.20 µg/mL.

**Table 1 ijms-25-07454-t001:** Ingredients reported on the label of the face cream. Dosages were not available.

Cyclopentasiloxane
Dimethicone crosspolymer
Tocophenryl acetate
Stearyl glycyrrhetinate
BHT
Phenoxyethanol

## Data Availability

Data are available from the corresponding author C.S. on request.

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
