# Peer review of "Pseudohyperaldosteronism Due to Licorice: A Practice-Based Learning from a Case Series"

_ijms, 2024, doi:10.3390/ijms25137454_

Round 1

Reviewer 1 Report

Comments and Suggestions for Authors

The paper “Pseudohyperaldosteronism due to licorice: a practice-based learning from a case series” contributes to the growth of literature for research, physicians and food producers, especially nutrients with licorice.

Before  the manuscript acceptation for publication in “International Journal of Molecular Sciences”, the following items should be revised:

Introduction

The description of the aim is not specific.

Results

 The sentence “Collecting the personal history, the patient reported an important intake of licorice candies (about 120 g/day) before hospitalization.”

What was the time period for the consumption of these candies?

The sentence “Patient firstly denied the consumption of licorice, grapefruit juice, or glucocorticoid therapy, but he then recalled the habit of drinking licorice liqueur at bedtime every day.” 

what was the concentration of lycorine in licorice liqueur?

Whether the authors determined the content of liquorice in the products consumed?

 Discussion

The sentence: “Finally, the toxic effects are influenced not only by the amount

of licorice intake, but also by several individual factors, such as long term use, female sex, aging, constipation, associated drug used, and comorbidities (as impaired cardiac, renal or liver function, and hypertension) [13].”

Did the authors find publications on what drug?

The Conclusions are extensive, similar to a summary, and missing two summarising sentences. What are the positive effects of the research? What are this research's limitations?

Author Response

COMMENT 1:

The paper “Pseudohyperaldosteronism due to licorice: a practice-based learning from a case series” contributes to the growth of literature for research, physicians and food producers, especially nutrients with licorice. Before  the manuscript acceptation for publication in “International Journal of Molecular Sciences”, the following items should be revised:

Introduction. The description of the aim is not specific.

RESPONSE 1: Thank you for your detailed revision that has significantly improved our paper. We have revised the introduction and better explained the aim of this paper.

COMMENT 2:

Results. The sentence “Collecting the personal history, the patient reported an important intake of licorice candies (about 120 g/day) before hospitalization.” What was the time period for the consumption of these candies?

RESPONSE 2: we have added the period for the consumption of licorice candies ( two weeks prior to hospitalization).

COMMENT 3:

The sentence “Patient firstly denied the consumption of licorice, grapefruit juice, or glucocorticoid therapy, but he then recalled the habit of drinking licorice liqueur at bedtime every day.” what was the concentration of lycorine in licorice liqueur? Whether the authors determined the content of liquorice in the products consumed?

RESPONSE 3: unfortunately it was not possible to evaluate the concentration of licorice in the liqueur consumed by the patient. Moreover, concetration of licorice varies among different beverages and it may be not well specified. We have added this aspect in the paper.

COMMENT 4:

Discussion. The sentence: “Finally, the toxic effects are influenced not only by the amount of licorice intake, but also by several individual factors, such as long term use, female sex, aging, constipation, associated drug used, and comorbidities (as impaired cardiac, renal or liver function, and hypertension) [13].” Did the authors find publications on what drug?

RESPONSE 4: Thank you for this important annotation. We have added in the text some drugs that may increase licorice toxicity.

COMMENT 5:

The Conclusions are extensive, similar to a summary, and missing two summarising sentences. What are the positive effects of the research? What are this research's limitations?

RESPONSE 5: Thank you, we have rewritten the conclusions and added at the end of discussion strengths and weaknesses of the paper.

Reviewer 2 Report

Comments and Suggestions for Authors

This article evaluates the pseudohyperaldosteronism due to licorice. The topic is relevant, but the major deficiencies identified in both content and form need to be addressed based on the specific recommendations below:

1.    The concluding part of the abstract should be improved in terms of results and future   research directions to which this research can refer.

2. The first section should be called introduction, exactly as in the template provided by the journal.

3. While mentioning grapefruit abuse, its enzyme inhibitory effect should also be mentioned, which can influence the metabolism of drugs that are metabolized by CYP450.

4. The international Latin name of licorice should be introduced for better clarity.

5. As there are two consecutive sentences, do not put the bibliographic index until the end (see bibliographic index [7]).

6. The last paragraph of the introduction is intended for the purpose of the study, highlighting the novelty and contribution to the literature of the work proposed by the authors.

7. The title of the table is placed above it as in the instructions for authors provided by the journal which I suggest you review. Font and text size must be respected also from the title up to and including the references.

8.  The present paper cannot be classified as article type (no IMRAD structure is identified in the text), so the authors must decide in which form they want to present it, either review, where they will have to detail much more the pathology, the licorice benefits (phytochemical composition, pharmacological action, combinations, results of in vitro and in vivo studies, etc.), or present it as a case report, but then the information should be presented more clearly (smaller and more detailed paragraphs, respect the structure of a case report, see other examples in this journal) and the novelty brought to the literature should be better highlighted.

9. The bibliographical resources used are mostly quite old, more than 5 years, so it is necessary to review the latest publications on this topic and adapt them to the present work.

Author Response

This article evaluates the pseudohyperaldosteronism due to licorice. The topic is relevant, but the major deficiencies identified in both content and form need to be addressed based on the specific recommendations below:

  1.  The concluding part of the abstract should be improved in terms of results and future   research directions to which this research can refer.

Thank you for your revision that have significantly improved the paper.

We have rewritten the final part of the abstract.

  1. The first section should be called introduction, exactly as in the template provided by the journal.

Done

  1. While mentioning grapefruit abuse, its enzyme inhibitory effect should also be mentioned, which can influence the metabolism of drugs that are metabolized by CYP450.

We have specified this important mechanism related to grapefruit.

  1. The international Latin name of licorice should be introduced for better clarity.

Done.

  1. As there are two consecutive sentences, do not put the bibliographic index until the end (see bibliographic index [7]).

Done.

  1. The last paragraph of the introduction is intended for the purpose of the study, highlighting the novelty and contribution to the literature of the work proposed by the authors.

We have revised the paper.

  1. The title of the table is placed above it as in the instructions for authors provided by the journal which I suggest you review. Font and text size must be respected also from the title up to and including the references.

Done.

  1. The present paper cannot be classified as article type (no IMRAD structure is identified in the text), so the authors must decide in which form they want to present it, either review, where they will have to detail much more the pathology, the licorice benefits (phytochemical composition, pharmacological action, combinations, results of in vitro and in vivo studies, etc.), or present it as a case report, but then the information should be presented more clearly (smaller and more detailed paragraphs, respect the structure of a case report, see other examples in this journal) and the novelty brought to the literature should be better highlighted.

As suggested even by the editor, the paper has been changed to case report.

9. The bibliographical resources used are mostly quite old, more than 5 years, so it is necessary to review the latest publications on this topic and adapt them to the present work.

More recent references have been added.

Round 2

Reviewer 2 Report

Comments and Suggestions for Authors

The authors have significantly improved the manuscript based on the suggestions received.